# Traditional, Vegetarian, or Low FODMAP Diets and Their Relation to Symptoms of Eating Disorders: A Cross-Sectional Study among Young Women in Poland

**DOI:** 10.3390/nu14194125

**Published:** 2022-10-04

**Authors:** Weronika Gwioździk, Karolina Krupa-Kotara, Beata Całyniuk, Paulina Helisz, Mateusz Grajek, Joanna Głogowska-Ligus

**Affiliations:** 1Department of Epidemiology, Faculty of Health Sciences in Bytom, Medical University of Silesia in Katowice, 41-902 Bytom, Poland; 2Department of Human Nutrition, Faculty of Health Sciences in Bytom, Medical University of Silesia in Katowice, 41-808 Zabrze, Poland; 3Department of Public Health, Faculty of Health Sciences in Bytom, Medical University of Silesia in Katowice, 41-902 Bytom, Poland

**Keywords:** eating disorders, orthorexia, diet therapy, vegetarianism, low FODMAP

## Abstract

Background: Eating disorders (EDs) are an ever-growing problem, seen most commonly among young women. People with EDs are particularly predisposed to restrictive eating patterns. Diet therapy for many diseases involves the elimination of certain food groups, and the use of restrictive diets in people with EDs may indirectly influence the recurrence of the disorder. The aim of this study was to assess the prevalence of the possible presence of EDs and orthorexia symptoms among young women following a traditional, alternative (vegetarian), or therapeutic (low FODMAP) diet. Methods: The analysis was based on 420 responses from female respondents aged 19 to 30 years. The research tool consisted of questionnaires to assess the possible presence of EDs (SCOFF), cognitive-behavioral aspects of eating (TFEQ-13), and the presence of orthorexia symptoms (ORTO-15). Results: Uncontrolled and emotional eating was least common in women on a vegetarian diet (5.6 ± 3.7; 3.5 ± 2.7; *p* < 0.05). Women on a low FODMAP diet were most prevalent with symptoms of orthorexia (*n* = 37, 47.4%; *p* < 0.05). Conclusions: It seems important to perform screening tests for eating disorders before recommending diet therapy in order to individually adjust the dietary regimen and refer patients to appropriate specialists.

## 1. Introduction

Advances in technology, resulting in increased opportunities for constant contact between people from all over the world, the rapid transport of goods, and the movement of people, mean that the public’s knowledge of nutrition and the ability to modify their diets with different products to suit their needs is increasing [1]. The topic of nutrition is becoming an increasingly popular and fundamental part of caring for one’s health, with the result that the health benefits of alternative diets are being recognized. At the same time, the high pace of life increasing perceived stress levels, the prevalence of learning and working remotely as a result of the COVID-19 pandemic, and the popularity of social media, which often has a significant negative impact on the perception of one’s own body, encourage the development of eating disorders and erroneous dietary choices, to which young women are especially vulnerable [2,3].

The basic diet of the population is called the traditional or habitual diet. It may vary according to the environment, family traditions, and individual habits but does not imply any special exclusions, such as the elimination of meat, dairy or cereal products. The traditional diet, depending on its energy value and the ratio of individual macronutrients and micronutrients, can be successfully used in the process of normalizing body weight and maintaining normal body function.

In situations where the state of health requires a targeted modification of the diet in terms of the supply of specific nutrients or a change in the consistency of meals, a therapeutic diet is used. Its purpose is to protect the diseased organ and to replenish micronutrients that have become deficient as a result of the disease. The healing diet must not contain any ingredients to which the patient is allergic or whose metabolism in the body is restricted. Culinary techniques and the range of foods used may also be modified. The changes made are necessary for the proper functioning of the individual patient’s organism and his or her quality of life, and the duration of the therapeutic diet depends, among other things, on the severity of the disease and co-morbidities [4]. Therapeutic diets can include various types of elimination diets, e.g., gluten-free in the case of coeliac disease, dairy-free in people with an allergy to cow’s milk proteins, or ketogenic when used as a result of medical indications (e.g., in drug-resistant epilepsy) [5].

The low FODMAP protocol is an example of a therapeutic diet used in patients with Irritable Bowel Syndrome (IBS) and Small Intestinal Bacterial Overgrowth (SIBO) [6]. The name of the diet is an acronym for Fermentable Oligosaccharides, Disaccharides, Monosaccharides, and Polyols, and its basic premise is the elimination of products containing the above-mentioned carbohydrates. The diet follows two stages: elimination and reintroduction. In the first stage, lasting 4 to 6 weeks, foods containing FODMAPs are completely excluded, including lactose-containing dairy products, high fructose fruits, products high in galactooligosaccharides (GOS) or fructooligosaccharides (FOS) (Table 1) [7,8].

In the second stage, the diet is expanded by gradually introducing previously excluded foods by the source of individual fermentable carbohydrates. A key element of the second stage is self-observation to determine individual tolerance to particular foods. This allows the traditional diet to be adapted to the needs of the individual patient once the low FODMAP diet has been completed [7].

In addition to therapeutic diets aimed at people who are suffering from illnesses, there are a number of alternative diets that can be followed by healthy people. Vegetarianism is constantly becoming more and more popular around the world and in recent years also in Poland. The vegetarian diet belongs to the group of alternative diets, and its main premise is the exclusion of meat. Heterogeneous motivations for vegetarians to follow this dietary trend have been pointed out, and these include both ethical and health reasons [8]. There are many types of vegetarian diets, and their division is based on the categories of products allowed or excluded from the diet. Diets with a high supply of plant-based products, such as vegetables, whole-grain cereal products, dry pulses, or fruit, have a positive impact on health, and a well-balanced vegetarian menu shows the potential to reduce the risk of cardiovascular disease, obesity, or type 2 diabetes. The reason for this is the high supply of dietary fiber, bioactive compounds, and micro- and macronutrients while keeping saturated fat and simple sugars low. Furthermore, in addition to ethical considerations, the popularity of vegetarianism is justified by its beneficial impact on the environment, as diets containing lower amounts of animal products show a lower carbon footprint [9,10].

Eating disorders (EDs) combine psychological, bodily, and health problems, and their etiology consists of many different aspects, among which are the role of childhood, broad self-regulation, personality, genetics, society, and culture [11,12]. According to the definition found in the International Classification of Diseases 11th Revision (ICD-11), eating disorders are abnormal eating behaviors that are not due to other disorders (psychiatric, neurodevelopmental or behavioral), medical conditions, or cultural conditions and are not developmentally appropriate. In addition, EDs include excessive preoccupation with food and heightened concerns about weight and body image [13]. Globally, the prevalence of eating disorders is estimated to have increased from 3.5 to 7.8% between 2000 and 2018 [14], and in the United States alone, the total economic costs associated with treating EDs, their psychiatric or somatic complications amounted to nearly $65 billion in the 2018–2019 fiscal year [15]. Although these disorders are most often diagnosed in young girls and mainly affect Western cultures, they can occur in people of any gender, age, and background [11]. Recent years have been particularly challenging for people with eating disorders due to the COVID-19 pandemic, as restrictions on social life and difficult access to medical professionals may have negatively affected their mental health [16,17]. It was observed that in 2020, rates of eating disorder symptoms, depression, and anxiety were higher than in previous years [18]. Due to possible somatic and psychiatric complications and the increasing prevalence of eating disorders, EDs are a serious public health problem. Eating disorders are diagnosed on the basis of the ICD-11 or DSM-5 classification. According to the ICD-11, eating disorders include anorexia nervosa (6B80), bulimia nervosa (6B81), binge eating disorder (6B81), and avoidant-restrictive food intake disorder (ARFID; 6B83). For abnormal eating behaviors that do not meet the diagnostic criteria for the above disorders, a diagnosis coded as 6B8Y, or other specific feeding or eating disorders, may be appropriate. These include, among others, orthorexia nervosa, which is not a separate disease entity but is seen by some to belong to the spectrum of eating disorders [13,19].

Orthorexia nervosa is a relatively new form of obsessive–compulsive eating that is described as a pathological obsession with healthy eating. It belongs to the category of eating-related disorders, but because of its psychosomatic and social dysfunctional problems, and at the same time, because of the magnitude of the dietary restrictions, the diagnostic criteria and the exact definition of orthorexia nervosa are still subject to scientific debate [20,21]. The study of orthorexia nervosa and its treatment is of particular importance because among the consequences of this putative disorder are nutritional deficiencies, malnutrition, underweight, and a variety of health-related consequences, even leading to death [22]. The main reason for the development of orthorexia nervosa is individual psychological conditions, but the widespread public interest in healthy eating over the past decade as a major contributor to the development of lifestyle diseases may also have an impact. Although sound health and nutrition education is welcomed by public health authorities, the information provided in the mass media and the popularity of alternative diets have an impact on the development of sometimes misconceptions about healthy eating and its impact on health, as well as the development of an excessive focus on so-called ‘clean’ eating. The above factors can lead to a pathological fixation on food and lead to negative health consequences (Figure 1) [23,24].

Given the above, the study aimed to assess cognitive-behavioral aspects of eating and the possibility of the occurrence of eating disorders and orthorexia symptoms among young women of reproductive age following a specific type of diet: traditional, alternative (vegetarian), or therapeutic (low FODMAP).

Based on the set main goal, the following specific goals can be formulated:
Assessment of the potential prevalence of eating disorders and orthorexia symptoms and analysis of cognitive-behavioral eating behaviors among women aged 19–30 years.Analysis of the correlation between the potential prevalence of eating disorders and orthorexia symptoms and the type of diet followed.Analysis of the correlation between cognitive-behavioral aspects of eating and type of diet followed.

The following working hypotheses were used in the study design:Eating disorder symptoms are common in young women.Women on elimination diets tend to restrict food more and are more likely to have symptoms of eating disorders and orthorexia.Emotional and uncontrolled eating is correlated with the type of diet followed.

## 2. Materials and Methods

### 2.1. Study Organization and Eligibility Criteria

The study was conducted in three stages. The first stage was a pilot study, during which 30 randomly selected women were asked to complete a questionnaire to check whether all questions were understandable. The majority of questions were found to be clear and understandable by the respondents, while questions that were indicated by at least 2 respondents as not understandable or unclear were removed or framed. Stage two was questionnaire validation by distributing the questionnaires twice to a randomly selected group of 30 women. An interval of 2 weeks was maintained between the collection of the questionnaires. The responses to the same questions were checked for consistency. To assess the reproducibility of the results obtained with the used questionnaire, the value of the parameter κ (Kappa) was calculated for each question in the questionnaire—for 61.3% of the questions, a very good (κ ≥ 0.80) concordance of answers was obtained, while for 31.7% of the questions, a good (0.79 ≥ κ ≥ 0.60) concordance of methods was obtained. The final stage of the study was to conduct the actual test.

Due to the ongoing COVID-19 pandemic, the survey was distributed using the CAWI (computer-assisted web interview) method, being a way of collecting data and information in which the respondent completes questionnaires electronically. The survey was distributed by the authors of the study to women in various groups through the social networking site Facebook. In order to avoid the phenomenon of bot/random responders, the necessity of giving answers to all questions was applied; moreover, different types of questions (closed, open, and semi-open) were used. In important aspects, the method of cross-questioning was used, which excluded the possible preparation of data by bots. In addition, the survey was available in thematic Facebook forums to which only users approved by the administrator had access; finally, after completing the survey, it was necessary to provide an alphanumeric or image code CAPTCHA. The general survey took place one month after the piloting to avoid the freshness effect, that is, the tendency for the respondent to better remember those possible answers from the questionnaire that are in the final positions in the cafeteria. As a result, the respondent chooses more often among those answers that are at the end of the cafeteria.

The exclusion criteria for the study were age below 19 years and above 30 years, male gender, and use of a diet other than traditional, vegetarian, or low FODMAP. The survey was conducted among 509 respondents. After excluding respondents’ answers that did not meet the eligibility conditions for the study (female gender, age between 19 and 30 years, following a traditional, vegetarian, or low FODMAP diet), the analysis was based on responses from 420 questionnaires. Respondents’ answers were divided according to the type of diet used: traditional, vegetarian, and low FODMAP. The study was conducted between October 2021 and March 2022. The study remains consistent with the provisions of the Declaration of Helsinki. The study in light of the Act of 5 December 1996 on the professions of physician and dentist (Journal of Laws of 2011, No. 277, item 1634, as amended) is not a medical experiment. It was approved by the Bioethics Committee of the Silesian Medical University in Katowice (ID. PCN/CBN/0052/KB/127/22).

### 2.2. Study Procedure and Research Tool

The research tool was an anonymous questionnaire consisting of five parts: a personal questionnaire, the author’s questionnaire on diet and eating habits, the SCOFF questionnaire by Morgan et al. [25], the TFEQ-13 questionnaire in the Polish adaptation by Anna Dzielska et al. [26] and the ORTO-15 questionnaire by Donini et al. [27].

Demographic information such as age, gender, and education was collected. Anthropometric measurements reported by the subjects included height and current body weight. From these data, BMI was calculated and interpreted according to World Health Organisation recommendations. The breakdown by body mass index was as follows: underweight <18 kg/m^2^; normal 18–24.99 kg/m^2^; overweight 25–30 kg/m^2^; obese >30 kg/m^2^ [28]. Health status questions asked about the presence of diet-related diseases that may affect eating behavior.

The SCOFF questionnaire was used to assess the prevalence of the possible presence of eating disorders. It consists of five questions addressing the basic characteristics of disorders such as anorexia nervosa and bulimia nervosa. Possible answers are ‘yes’ or ‘no’. Giving two or more affirmative answers indicates the presence of an eating disorder. It is a simple screening tool that is not used for diagnosis but to suggest the likelihood of an eating disorder in a given case [25].

The Polish adaptation of the Three-Factor Eating Questionnaire (TFEQ-13) was used to assess cognitive-behavioral aspects related to eating. In the original version by Stunkard and Messik, the TFEQ scale contained 51 questions. It was shortened to 18 questions by Karlsson and then translated and reduced on the basis of testing. In the final version used for the study, the TFEQ-13 scale contains 13 questions, which are divided into three subscales: restrictive eating (control over food intake to control body weight; R1–R5), uncontrolled eating (general difficulties in regulating eating; J1–J5) and emotional eating (overeating in depressed mood states; E1–E3). Responses to each question were standardized on a 4-point scale and interpreted by scoring from 0 to 3 (definitely no—0 points, rather no—1 point, rather yes—2 points, definitely yes—3 points). Question 13, which involves indicating a number on a scale from 0 to 8, is coded differently: values 1 and 2 are 0 points, 3 and 4 are 1 point, 5 and 6 are 2 points, and 7 and 8 are 3 points. Values were calculated separately for each component, and the higher the score on a subscale, the greater the severity of impairment in that area [26,29,30].

Another questionnaire used for the study was the ORTO-15, which is the most commonly used measure in screening for symptoms of orthorexia. The questionnaire consists of 15 questions, and responses were created on a 4-point Likert-type scale, including statements of ‘always’, ‘often’, ‘rarely’, or ‘never’. The scoring of questions numbered 3, 4, 6, 7, 10, 11, 12, 14, and 15 is as follows: ‘always’ is 1 point, ‘often’ is 2 points, ‘rarely’ is 3 points, and ‘never’ is 4 points. For questions numbered 2, 5, 8, and 9: ‘always’ is 4 points, ‘often’ is 3 points, ‘rarely’ is 2 points, and ‘never’ is 1 point. For questions numbered 1 and 13: ‘always’ is 2 points, ‘often’ is 4 points, ‘rarely’ is 3 points, and ‘never’ is 1 point.

As suggested by the authors of the questionnaire, a score below 40 points indicates the presence of orthorexia symptoms [27], while the Polish validation of the questionnaire proposed a cut-off point of 35 points, suggested by the nature of the distribution of the orthorexia risk index in the study population [31]. The study used the interpretation that a score below 35 points indicates the presence of orthorexia symptoms.

### 2.3. Statistical Compilation

The database was compiled using Microsoft Excel, while statistical analysis was performed using Statistica 13.3 software (TIBCO Software Inc., Palo Alto, CA, USA). The values of measurable parameters that were analyzed were presented by mean value and standard deviation, while non-measurable parameters were presented by counts and percentages. To identify differences between groups, qualitative independent characteristics were analyzed by the Chi^2^ test of homogeneity and quantitative characteristics by Kruskal–Wallis rank analysis of variance. The level of statistical significance was considered to be *p* < 0.05.

## 3. Results

In order to determine the characteristics of the study group, the responses of the female respondents were classified according to such parameters as age, BMI, and education. The average age of the respondents was 24 years. The body mass index of the majority of the women surveyed was within the normal range (*n* = 273; 65.0%). The respondents most often declared that they had a college education or were pursuing a degree (*n* = 370, 88.1%), with more than 1/4 of the respondents indicating medical and health sciences as their field of education (*n* = 113, 26.9%). The women surveyed were assigned with respect to the type of diet used. More than half of the respondents followed a traditional diet (*n* = 227, 54.1%), a vegetarian diet was followed by 27.4% (*n* = 115), and a low FODMAP diet was followed by 18.6% (*n* = 78) (Table 2). The difference in the size of the study groups by type of diet corresponds to the frequency of use of the given dietary patterns in the general population.

Most of the women surveyed had been on a traditional or vegetarian diet for longer than 2 years (*n* = 130, 57.3%; *n* = 73, 63.5%). Respondents on a low FODMAP diet mostly followed it for less than a year, which is in line with the assumptions of this treatment protocol (*n* = 57, 73.1%). The difference in the duration of use of the respective types of diets among the respondents was statistically significant (*p* < 0.05; Table 3). Almost half of the women surveyed had diet-related diseases (*n* = 194, 46.2%). The most frequently cited included inflammatory bowel disease, SIBO, insulin resistance, and diabetes. Almost all of the respondents who followed a low FODMAP diet had diet-related diseases (*n* = 75, 96.2%), indicating the correct use of a therapeutic diet. The lowest percentage of respondents who had diet-related diseases was observed among those who followed a vegetarian diet (*n* = 34, 29.6%). The difference in the prevalence of the aforementioned conditions in the groups of women following traditional, vegetarian, and low FODMAP diets was statistically significant (*p* < 0.05). The results of the respondents’ answers with percentages are presented in Table 3.

The women surveyed were asked to choose the main reason why they follow a particular type of diet. Respondents on a traditional diet most often indicated responses of “I’ve been eating this way forever” (*n* = 97, 42.7%) and “I think it’s the best choice for my health” (*n* = 57, 25.1%). Vegetarians also suggested a positive impact of their diet on their health (*n* = 62, 53.9%), and one in five cited ethical, environmental, and worldview motivations (*n* = 23, 20%). The low FODMAP diet was most often followed on the recommendation of a dietitian (*n* = 25, 32.1%) or doctor (*n* = 23, 29.5%), consistent with the therapeutic nature of this dietary regimen. Other reasons cited by respondents included lack of time, money, and energy to change their diet (in the case of the traditional diet) and reluctance to eat meat due to its taste (in the case of vegetarianism).

The women surveyed were asked about their use of the dietitian consultation. Most of them had never reached out for professional dietary advice (*n* = 245, 58.3%), and 8.3% were under the regular care of a dietitian (*n* = 35, 8.3%). The surveyed women who followed a low FODMAP diet were most likely to have used a dietitian in the past or currently (*n* = 56, 71.8%).

### 3.1. Results of the SCOFF Questionnaire

Analysis of the SCOFF questionnaire showed that more than half of the women surveyed may have symptoms of an eating disorder (*n* = 243, 57.9%). Respondents most often answered positively to the questions “Do you worry you have lost control over how much you eat?” (*n* = 208, 49.5%) and “Would you say that food dominates your life?” (*n* = 224, 53,3%). More than 40% of respondents perceived themselves to be overweight (*n* = 183, 43.8%). No significant statistical differences were found between the possible occurrence of eating disorder symptoms and the type of diet used (Table 4).

### 3.2. Results of the ORTO-15 Questionnaire

In order to assess the possible occurrence of orthorexia symptoms in the study group, the scores obtained on the ORTO-15 questionnaire completed by the female respondents were calculated. The lowest possible score was 15 points, and the highest was 60 points. The mean point value of the respondents’ answers was 36.3 ± 3.8 points, with a score of more than 35 points considered the cut-off point for determining the presence of orthorexia symptoms. The analysis showed that one in three respondents had symptoms of orthorexia (*n* = 129, 30.7%).

The results of the ORTO-15 questionnaire were divided according to the type of dietary regimen used by the subjects. Statistical analysis showed significant statistical differences between the type of diet used and the occurrence of orthorexia symptoms. The group of subjects following the low FODMAP diet had the highest frequency of the occurrence of orthorexia symptoms (*n* = 37, 47.4%, *p* < 0.05) (Table 5).

### 3.3. Cognitive-Behavioral Aspects of Eating of Female Respondents

Responses to the TFEQ-13 questionnaire allowed analysis of the psychological aspects of eating of the women surveyed, with a breakdown into restrictive eating, lack of control over eating, and eating under the psychological influence. The respondents achieved the highest score in the subscale of restrictive eating (6.7 ± 3.6). On the uncontrolled eating subscale, respondents scored an average of 6.4 ± 3.7, while on the emotional eating subscale, the average score was 4.0 ± 2.7.

Differences in psychological aspects of eating between groups of female subjects on traditional, vegetarian, or low FODMAP diets were analyzed. The highest scores for uncontrolled and emotional eating were observed among women eating traditionally (6.8 ± 3.5; 4.4 ± 2.7; *p* < 0.05). Respondents following a low FODMAP diet had a slightly higher score on the restrictive eating subscale (7.0 ± 3.4; *p* > 0.05). The mean scores for each emotional subscale within the respective study groups are shown in Table 6.

## 4. Discussion

Health, according to its modern, holistic conception, is a state in which a person can function properly physically, mentally, socially, and spiritually [32]. Internal balance and well-being can be disrupted by, among other things, eating disorders. In our study, no correlation was observed between adherence to a vegetarian diet and the possible occurrence of eating disorders; moreover, vegetarians had the lowest scores for the emotional and uncontrolled eating subscales. In a study by Heiss et al. [33], vegetarians were found to be less likely to exhibit symptoms of an eating disorder than those not using any restriction or elimination, and Norwood et al. observed that the prevalence of behaviors suggestive of an eating disorder was lower in the vegan group than in other groups. In addition, they showed less tendency to eat emotionally and less motivation to control weight [34], which also coincides with the results of our study. Dorard and Mathieu [35] showed that women on a plant-based diet were characterized by lower body and weight concerns and were less likely to focus on controlling their weight than women on a traditional diet. Given the motivation of vegetarians to eat naturally and healthily, the authors noted the need to screen for orthorexia among this group [35]. In our study, vegetarian women were slightly more likely to show symptoms of orthorexia than women following a traditional diet. These findings are supported by a review of the literature by Brytek-Matera [36], where it was shown that following a vegetarian diet is associated with orthorexic eating behavior. However, the author of the study points out the two-dimensionality of orthorexia nervosa, indicating the existence of healthy orthorexia, resulting from a non-pathological interest in healthy eating, and nervous orthorexia, associated with a pathological preoccupation with healthy eating. Therefore, it is pointed out that these two dimensions need to be distinguished among those following a vegetarian diet in further studies [36]. The analysis by Dittfeld et al. [37] also found that excessive preoccupation with healthy food is characteristic of vegetarians compared to those on a traditional diet but that the risk of orthorexia decreases with age and diet duration. It is also pointed out that vegetarianism can be used to hide eating disorders, as it allows sufferers to avoid certain food groups or food-related situations. On the other hand, the experience of following a plant-based diet may increase the risk of developing eating disorders in people who are particularly predisposed based on personality factors [38]. Therefore, a cause-and-effect relationship for eating disorders in vegetarians is still being sought [37].

A vegetarian diet, as a way of eating assuming certain restrictions, is most often chosen voluntarily. The situation is different for therapeutic diets, of which the low FODMAP protocol is an example. In our study, this diet was most often followed on the recommendation of a dietician or doctor. Similar results were obtained by Tuck et al. [39], where the low FODMAP diet was recommended in 53% of the subjects by a gastroenterologist, 22% by a general practitioner, and 9% by a dietician. One in three respondents had received professional dietary advice. Proper dietary adherence, which allowed therapeutic intake of FODMAPs (<12 g/d), was more often achieved by subjects educated by a dietitian [39]. In our study, one in four respondents on a low FODMAP diet was under the regular care of this professional, and almost half had used a professional nutrition service in the past. Bellini et al. [40] point out that a low FODMAP diet must be prescribed by healthcare professionals (such as a gastroenterologist or dietician) who have the appropriate nutritional knowledge and skills in this area, as the use of elimination diets without professional advice is associated with an increased risk of nutrient deficiencies and the development of eating disorders in predisposed patients [40].

Although the use of a low FODMAP diet helps to alleviate intestinal symptoms, it is associated with a number of burdens, such as medical costs or problems implementing the principles of this diet on a daily basis. Due to these inconveniences, this diet may even impair quality of life [41]. In addition, patients with gastrointestinal symptoms are observed to have abnormal eating behaviors, such as food avoidance and associated reduction of daily energy intake, irregular meal consumption, or skipping meals altogether. Reasons may include increased symptoms, fear of pain, and psychological stress. Although these behaviors may resemble traditional eating disorders in IBS patients, the difference lies in the motivation for engaging in them—the reason is not dissatisfaction with body weight but a desire to alleviate symptoms. However, if additional predisposing factors for the eating disorder are present or the maladaptive behavior is particularly severe, the risk of developing anorexia, bulimia, orthorexia, or ARFID increases [42]. It is particularly dangerous to categorize foods as ‘good and bad’ or ‘healthy and unhealthy’, as patients may overestimate the importance of food for symptom severity and reinforce beliefs about the harmfulness of certain foods—this should be noted especially among patients with orthorexia [42]. In our study, the presence of orthorexia symptoms was observed in almost half of the subjects following a low FODMAP diet, which was the highest percentage among all the women studied. Although no significant differences were observed in the incidence of eating disorders among subjects on the low FODMAP diet, the topic of potential links between the incidence of EDs and gastroenterological problems is worth considering. High preoccupation with nutrition and diet, anxiety related to the severity of symptoms, and particular attention to body signals and appearance among women with intestinal complaints may be significant factors that increase the possibility of eating disorders. It is difficult to assess the causal sequence in the association of eating disorders with the use of a low FODMAP diet, as it is noteworthy that people diagnosed with eating disorders very often face gastrointestinal problems that are associated with altered gut-brain interaction in the absence of structural abnormalities detectable at the diagnostic stage [43]. The behaviors undertaken by individuals with eating disorders may negatively affect the functioning of the gastrointestinal tract and the composition of the gut microbiota, thereby compounding the dysfunction of the gut–brain axis. On the other hand, gastrointestinal complaints, due to the previously mentioned factors, may precede the occurrence of eating disorders and predispose them to their development [44]. The coexistence of these diseases or misdiagnosis can lead to complications in treatment, compromising the patient’s quality of life. Therefore, clinicians working with patients suffering from somatic EDs are suggested to use screening questionnaires to detect ED risk, such as SCOFF, in order to be able to correctly plan treatment or refer the patient to an appropriate specialist (e.g., psychiatrist or psychologist). Riedlinger et al. also point out the need to raise awareness among psychotherapists working with people with eating disorders about the prevalence of gastrointestinal complaints in this patient group and possible screening for other gastrointestinal conditions [45]. Due to the possible recurrence or worsening of eating disorders during an elimination diet, in patients who have suffered from eating disorders, diet therapy with functional foods or brain and gut psychotherapy is suggested instead of following a restrictive dietary regimen [44,46].

## 5. Strengths and Study Limitations

The strengths of the study certainly include its innovation: the use of traditional, alternative, and therapeutic diets has not previously been compared in relation to eating disorder risk, orthorexia, and cognitive-behavioral aspects of eating. Also important is the focus on a comprehensive approach to a patient requiring a therapeutic diet but also suffering from an eating disorder.

Undoubtedly, the strength of the study conducted is the large group of 420 subjects selected from a group of 509. Taking into account data from the Central Statistical Office in Poland, according to which the population of women aged 19–30 is approximately 2,143,014, the study group fulfills the conditions for minimum sampling [47]. The distribution of the study group in relation to the diet used corresponds to the prevalence of the diet use in Poland: according to the Ariadna Panel study, 8.4% of Poles declare vegetarianism [48]. Approximately 11% of the population suffers from irritable bowel syndrome, but unfortunately, there are no reliable statistical data on the use of low FODMAP diets in this group [49].

Conducting a pilot study and calculating Kappa statistics also adds to the value of the study. However, the survey is not free from methodological limitations; conducting surveys using the CAWI method does not avoid the common phenomenon of “bot/fakeresponders”, which is characterized by surveys made available on the basis of online forms; however, all possible means were used to avoid them; Nevertheless, in the case of the tool used, we have a high degree of confidence that the questionnaires were filled out reliably, which was also confirmed during the pilot study and validation of the survey.

Unfortunately, despite the desire to use the most reliable questionnaires possible, one questionnaire has been updated—a more recent tool than the ORTO-15 is the ORTO-R. Nevertheless, the ORTO-15 was validated in Poland, and this questionnaire was used in the study [50]. Given that this is a cross-sectional study based on the subjective assessment of the subjects, the results and conclusions should be treated as preliminary information, and further research should be conducted.

## 6. Conclusions

Women on the traditional diet had the highest scores for uncontrolled and emotional eating. In contrast, women on a vegetarian diet had the lowest scores for uncontrolled and emotional eating of all respondents. Women on the low FODMAP diet had a slightly higher score for restrictive eating than respondents on the vegetarian and traditional diets. Furthermore, this group had the highest percentage of orthorexia symptoms. The low FODMAP diet was most often used on the recommendation of a dietician or doctor, so it seems important to screen for eating disorders before recommending diet therapy in order to individually adjust the dietary regimen and refer patients to appropriate specialists.

## Figures and Tables

**Figure 1 nutrients-14-04125-f001:**
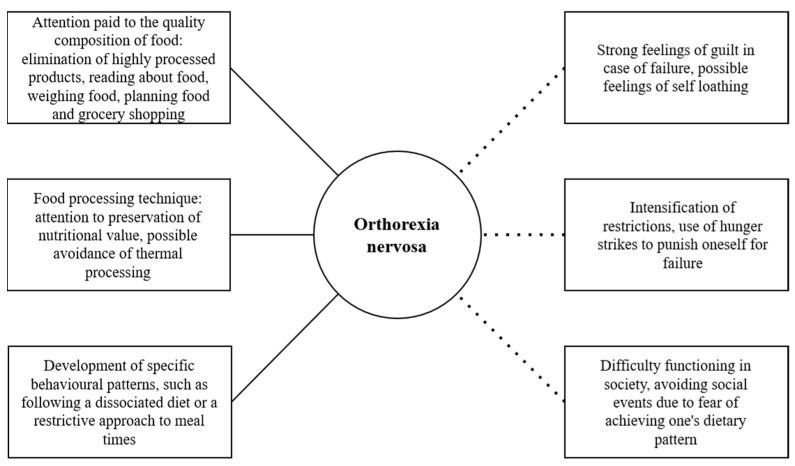
Characteristics in orthorexia nervosa.

**Table 1 nutrients-14-04125-t001:** Foods by type of FODMAP.

Food Group	Type of FODMAP	Examples of Products with High FODMAP Content
Fruits	Sorbitol	Blackberries, peaches, plums
	Fructose (high content)	Mangoes, cherries, figs, watermelons, dried fruit
	Sorbitol and fructose	Apples, pears, cherries
Vegetables	FOS ^1^	Artichoke, garlic, onion, chives
	Mannitol	Cauliflower, snow peas
Cereal products	FOS, rarely GOS ^2^	Wholemeal bread, rye bread, wheat pasta, rye crispbread, muesli containing wheat
Dried pulses	GOS	Red beans, peas, chickpeas
Dairy products	Lactose	Milk, soft cheeses, yogurts
Nuts and seeds	GOS, FOS	Cashew nuts, pistachios
Sugars and sweeteners	Fructose	Honey, corn syrup, sweeteners containing fructose
	Sugar alcohols	Xylitol, sorbitol, erythrol, sugar-free sweets

^1^ FOS—fructooligosaccharides; ^2^ GOS—galactooligosaccharides.

**Table 2 nutrients-14-04125-t002:** Socio-demographic data of surveyed women.

Socio-Demographic Data		*n*	% of *n*
Age	19–24 y.o.	235	56.0
	25–30 y.o.	185	44.1
BMI	Underweight	56	13.3
	Healthy weight	273	65.0
	Overweight	67	16.0
	Obesity	24	5.7
	Elementary	2	0.5
Education	Technical	6	1.4
	Secondary	42	10.0
	Higher/while studying	370	88.1
Diet	Traditional	227	54.1
	Vegetarian	115	27.4
	Low FODMAP	78	18.6

**Table 3 nutrients-14-04125-t003:** Distribution of respondents according to the type of diet with respect to the duration of its use.

Duration of Dietary Use	Traditional Diet	Vegetarian Diet	LOW FODMAP Diet
	*n* (%)	*n* (%)	*n* (%)
Less than one year	63 (27.8)	17 (14.8)	57 (73.1)
1–2 years	34 (15.0)	25 (21.7)	9 (11.5)
More than 2 years	130 (57.3)	73 (63.5)	12 (15.4)
Statistical analysis: Chi^2^ = 80.01; *p*-value = 0.000			
**Diet-related diseases**			
No	142 (62.6)	81 (70.4)	3 (3.9)
Yes	85 (37.4)	34 (29.6)	75 (96.2)
Statistical analysis: Chi^2^ = 80.01; *p*-value = 0.000			

**Table 4 nutrients-14-04125-t004:** Results of the SCOFF questionnaire in a group of female respondents.

Type of Diet vs. SCOFF Questionnaire Results	Probability of the Presence of an Eating Disorder	No Probability of the Presence of an Eating Disorder
	*n* (%)	*n* (%)
Traditional diet	126 (55.5)	101 (44.5)
Vegetarian diet	64 (55.7)	51 (44.4)
Low FODMAP diet	53 (68.0)	25 (32.1)
Total results	243 (57.9)	177 (42.1)
Statistical analysis: Chi^2^ = 4.00; *p*-value = 0.135		

**Table 5 nutrients-14-04125-t005:** Results of the ORTO-15 questionnaire in a group of female respondents.

Type of Diet vs. ORTO-15 Questionnaire Results	Probability of Having Symptoms of Orthorexia	No Probability of Having Symptoms of Orthorexia
	*n* (%)	*n* (%)
Traditional diet	54 (23.8)	173 (76.2)
Vegetarian diet	38 (33.0)	77 (67.0)
Low FODMAP diet	37 (47.4)	41 (52.6)
Total results	129 (30.7)	291 (69.3)
Statistical analysis: Chi^2^ = 15,66; *p*-value = 0.000		

**Table 6 nutrients-14-04125-t006:** Cognitive-behavioral aspects of eating based on TFEQ-13 vs. type of diet.

TFEQ-13 Results According to the Type of Diet Used	Restrictive Eating	Uncontrolled Eating	Emotional Eating
	Average ± SD ^1^	Average ± SD ^1^	Average ± SD ^1^
Traditional diet	6.5 ± 3.4	6.8 ± 3.5	4.4 ± 2.7
Vegetarian diet	6.7 ± 4.2	5.6 ± 3.7	3.5 ± 2.7
Low FODMAP diet	7.0 ± 3.4	6.3 ± 3.8	4.0 ± 2.7
*p*-value	*p* = 0.5808	*p* = 0.0096	*p* = 0.0105

^1^ Standard deviation.

## Data Availability

Not applicable.

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
