# Peer review of "Traditional, Vegetarian, or Low FODMAP Diets and Their Relation to Symptoms of Eating Disorders: A Cross-Sectional Study among Young Women in Poland"

_nutrients, 2022, doi:10.3390/nu14194125_

Round 1

Reviewer 1 Report

This is a useful contribution to a field of investigation that is increasingly of interest to researchers and practitioners in the eating disorders, nutrition and public health fields, namely, the association between different forms of dieting and the occurrence of eating-disordered behavior. Limitations of the study methods are such that the findings need to be interpreted with caution and any conclusions are necessarily tentative. Nevertheless, the research is likely to be of interest to most if not all readers of the journal in my view. The comments that follow can be taken as suggestions for how the manuscript might be improved as part of a revision, should this be the Editorial decision.

Title

Would it be preferable to replace “Low FODMAP” with “therapeutic”? I think this would make sense, given that many readers will not know what the acronym stands for until reading the Introduction.

Introduction

Please consider the use of more cautious language concerning the status of “orthorexia nervosa”, given concerns about this putative new eating disorder being an example of the pathologizing of normative eating behaviors (see the recent work of Ross-Arguedas and others, e.g., https://link.springer.com/article/10.1007/s11133-022-09518-2). For example, “… is seen by some to belong to the spectrum of eating disorders” rather than “certainly belongs to the spectrum of eating disorders”, “… form of obsessive-compulsive eating” rather than “obsessive-compulsive disorder”, “putative eating disorder/disorder” rather than "eating disorder/disorder”. As the authors’ note, the status of this condition remains unclear, hence it is important to avoid undue medicalization.

Materials and Methods

It is noted that “the survey was conducted among 509 respondents” and that “the analysis was based on responses from 420 questionnaires” but there is no information how/from where participants were recruited or why/how responses from 89 participants were deleted. This information needs to be included so that readers can assess (and the authors address) issues of potential sample bias/representativeness/generalizability. In the Discussion section it is noted that “Undoubtedly, the strength of the study is the large group of 420 subjects, selected from a group of 509, which was a representative group as far as studies conducted in these groups are concerned” but what evidence is there that the sample was in fact representative of young adult women in the Poland?

Has the SCOFF been validated in the current study population, i.e., young adult women in Poland? If not, and aside from any issues concerning the validity of this measure more generally, the finding (reported in the Results section) that “more than half of the women surveyed were at risk of having an eating disorder” needs to be interpreted with caution and this should be noted in the text addressing the study limitations.

Results

Numerous statistical tests were carried out yet there doesn’t appear to have been any adjustment to the alpha level in order to reduce the likelihood of false positives. Either the significance level should be adjusted to take into account the number of statistical tests or a rationale for not doing so should be stated. If, as seems likely, some findings will no longer be statistically significant when using a more stringent criterion, then the text of both the Results and Discussion sections (and Abstract) will need to be modified accordingly.

The number of tables (13!) is far too high for a relatively simple piece of research. I suggest that the authors review the Results section with a view to reducing the number of tables included to a maximum of 4. The information in some tables (e.g., Tables 2, 6, 7, 10 & 12) could either be incorporated in the text or simply deleted, while the information in others (e.g., Tables 3, 4, & 5; Tables 8 & 11) could be combined into a single table.

Discussion

The authors’ discussion of the implications of the study findings is a bit confusing. In the Abstract, it is noted that “There is a need to educate the public about obesity prevention factors and relaxation techniques to prevent overeating caused by psychological factors. It seems important to perform screening tests for eating disorders before recommending diet therapy”, while in the Discussion it is noted that “clinicians working with patients suffering from somatic EDs are suggested to use screening questionnaires to detect ED risk, such as SCOFF, in order to be able to correctly plan treatment or refer the patient to an appropriate specialist (e.g. psychiatrist or psychologist). Riedlinger et al. also point out the need to raise awareness among psychotherapists working with people with eating disorders about the prevalence of gastrointestinal complaints in this patient group and possible screening for other gastrointestinal conditions [45]”. It would be helpful if the implications of the findings for health promotion programs designed to improve “eating disorders mental health literacy” among individuals with symptoms, primary care practitioners, mental health professionals and the public were considered, particularly from a public health perspective and particularly given that the research was conducted in a general population sample (see in particular the recent review by Bullivant and colleagues: https://www.tandfonline.com/doi/abs/10.1080/09638237.2020.1713996).

The authors state that “Undoubtedly, the strength of the study conducted is the large group of 420 subjects, selected from a group of 509, which was a representative group as far as studies conducted in these groups are concerned”. As suggested above, in the absence of any evidence to support the claim of sample representativeness and given the use of self-report measures at least some of which do not appear to have been validated in this study population, the authors should be a bit more modest about putative “strengths of the study”. Further, the statement that “Using appropriately selected methods significantly reduced the researcher’s error” would better be deleted and the limitation inherent in the cross-sectional study design, namely that any inferences concerning the directions of the observed association are tentative at best, should be explicitly acknowledged. Given these limitations, the findings should be seen as providing preliminary information only and this should also be acknowledged.

Author Response

Dear Reviewer,
Thank you very much for reviewing our manuscript in detail and making many relevant comments for its improvement. We have tried to address them to the best of our ability. The changes made are indicated in blue in the lines: 
2
120-122
128
161-164
206-210
318
326-328
438-445
457-459

The number of tables as suggested have been reduced to 6, for the sake of readability of the results. The others have been included in the text.

Once again, we would like to thank you for the effort of undertaking the review.
With best regards, Authors

Reviewer 2 Report

I would like to congratulate the authors for this work, very current, pertinent and innovative. It is very well written, the methodology is very clear and the results are better described and studied. The discussion is well structured, I consider the excellent work - congratulations. 

I would just like to make some suggestions:
- line 46 refers to "individual habits and habits" I don't understand;
- Figure 1 was created by you or adapted from an author? if it has been adapted, put refª if it was designed by you congratulations.
- Line 206 justify why they chose the cut-off point 35,
- line 263 - thyroid disease is not a disease directly related to food... it can cause obesity/thinness but it does not have the same relationship with the diet as the others.
- the conclusion of the introduction should be improved based on the results they found.

Author Response

Dear Reviewer,
Thank you very much for your favorable review and for reviewing our manuscript in detail and providing comments to improve it. We have tried to address them to the best of our ability. The changes made have been highlighted in yellow in the lines: 

46
213-214

Once again, we would like to thank you for your efforts in undertaking the review.
With best regards, Authors